# Identification of Workplace Bullying: Reliability and Validity of Indonesian Version of the Negative Acts Questionnaire-Revised (NAQ-R)

**DOI:** 10.3390/ijerph18083985

**Published:** 2021-04-10

**Authors:** Dadan Erwandi, Abdul Kadir, Fatma Lestari

**Affiliations:** Occupational Health and Safety Department, Faculty of Public Health, Universitas Indonesia, Depok West Java 16424, Indonesia; abdul_kadir@ui.ac.id (A.K.); fatma@ui.ac.id (F.L.)

**Keywords:** bullying, workplace, Indonesian, NAQ-R, psychosocial distress, satisfaction with life

## Abstract

Bullying can pose a risk to health and safety, including the risk for damage to the emotional, psychosocial, mental, or physical health of employees in the workplace. Since bullying has a detrimental impact on victims and organizations, several studies on this issue had been conducted using the Negative Acts Questionnaire-Revised (NAQ-R), which is one of the most widely used tools to assess and minimize the occurrence of workplace bullying. However, this tool has not been validated for the Indonesian contexts. In this study, the author tested the reliability and constructed validity of the Indonesian version of NAQ-R. A total of 3140 participants were recruited in this study from various companies from different industries. NAQ-R, Psychosocial Distress (K10), and Satisfaction with Life Scale (SWLS) were administrated through an online survey. The results showed that 22 items yielded three model factors, i.e., person-related bullying, work-related bullying, and intimidation towards a person. Cronbach’s alpha coefficients for the total and sub-scales of the Indonesian NAQ-R was acceptable, ranging from 0.721 to 0.897. This study confirmed that the Indonesian version of NAQ-R has an internal consistency reliability, and the concurrent and construct validity are at acceptable levels. Thus, this tool can be used as the screening instrument in assessing workplace bullying.

## 1. Introduction

Bullying is a very damaging and dangerous complex and heterogeneous phenomenon that directly affects hundreds of million people per year. Understanding the definition of bullying has been proven to be massively useful as a starting point for research [1]. The most common definition of bullying is a repetition of a negative physical, verbal, or psychological action targeting a certain individual, which can be seen in both an organization and a community. It has been said that one of the origins of bullying is lessened community cohesion or the destruction of the warp and weft of the tapestries of society [2]. Interestingly, in its recent development, bullying has become one of the concerns in the Occupational Health and Safety (OHS) field. Bullying can happen to anyone in any workplace, regardless of rank or income-level, but it is more prevalent in particular professions, such as education and healthcare. The first identified study of bullying was a study performed in 1984 by a Swedish researcher, Heinz Leymann [3,4,5,6]. Here, bullying was defined as actions that involve exposure to either weekly or more frequent of at least one negative act or behavior in a minimum period of six months, which is also known as mobbing or psychological terror [4,7]. In addition, several other definitions of bullying have been proposed by researchers, including stalking or psychosocial measures, which are not always visible [8]. Moreover, Gupta [9] pointed out that bullying involves several behaviors such as torture, intimidation, undermining, and scaring the target person through physical, psychological, and emotional domination. Furthermore, bullying can also be understood as a situation that “refers to every hostile and aggressive actions subjected to at least one or more victims with stigma” [10]. Stigma is defined as an insidious process that creates inappropriate shame and unworthiness to the individual. Stigma can lead to bullying. When someone is being stigmatized, they will effectively cut off from the collective group. Moreover, bullies automatically deny access to resources, stability, protection, and social status for the stigmatized person. They set boundaries to prevent social movement by the person who is stigmatized, and the only membership available for this person would be in the outside group, where they may also be rejected due to the stigmatization [2]. Bullying and harassment are extremely similar in that they are often utilized conversely to represent harmful or negative behaviors to other persons. They, however, differ in terms of definition. Bullying tends to be prompted by a hidden personal characteristic of targets or victims, such as competence, popularity, or integrity. For example, work-related bullying may take the form of making unreasonable demands and taking credit for others’ work. In contrast, harassment refers to unwanted, offensive, and intrusive behaviors related to sexual, racial, or physical elements. In other words, harassment tends to use references to certain characteristics of an outward individual such as race, religion, gender, sexual-orientation, and disability and it has a strong physical component such as physical contact and damage to possessions [11].

### 1.1. Workplace Bullying

Bullying is not solely suffered by children in playgrounds. The number of cases of bullying has been dramatically rising in workplaces and has caused a detrimental effect on organizations. Many researchers have identified, in the last 20 years, related elements of this topic, such as the nature, antecedents, and consequences of workplace bullying [12,13,14]. Furthermore, various definitions have been suggested for bullying at work [15]. This phenomenon is defined as negative acts which can affect a person’s work task and involves harassment, offense, and social exclusion [16]. According to Branch [17], workplace bullying has been massively used to define negative workplace behaviors. The Workplace Bullying Institute (WBI) proposed the term of workplace bullying to represent repeated mistreatment and abusive acts [18]. Nonetheless, not all negative behaviors can be labeled as bullying [19,20]. Workplace bullying consists of three elements: regularity of incidents, consequences to both morale and health, and business standards regarding the treatment of employees [21]. Three models of workplace bullying have been clearly described. The first is work-related bullying, where a person withholds information that can affect performance. The second is person-related bullying, for instance actions related to persistent criticism of errors or mistakes and sarcasm. The third is bullying through physical intimidation, such as intimidating behaviors (finger-pointing, shoving, or blocking victim’s way). These models focus on the characteristic of the victim and perpetrator’s personality; the reaction of the organizational environment towards bullying; and human relationships [15,22].

### 1.2. Prevalence of Workplace Bullying

Investigations have been performed on the prevalence of workplace bullying around the world. In the United Kingdom, for instance, 10.6% of respondents in a random nationwide survey of 70 organizations were identified as bullying victims. According to a study by UNISON (the Public Service Union in the UK), workplace bullying has been identified in the public sector union with an incidence of 34% [23]. In addition, the 2017 report of WBI survey of the United State of America demonstrated that 19% of Americans have been subjected to abusive acts at the workplace, while another 19% have witnessed those actions and 63% have identified bullying at their workplace [18]. Moreover, several studies have reported the prevalence of bullying at work in different countries. The prevalence in European countries ranges between 3.5 and 10% [24], while the prevalence in New Zealand and Australia is 18% [25] and 25% to 50% [26,27], respectively. In India, a study reported that 53% of men and 35% of women were subjected to bullying, with 90% of bullying cases are not reported [28]. A cross-sectional Finnish study (1000 samples) showed that 11.5% of females become a victim of bullying as opposed to 5% of men [29].

### 1.3. Impact of Workplace Bullying

The effect of bullying on several aspects of life is now well-established. The impact of workplace bullying includes negative impacts on the bullying target as an individual, as well as on the organization, family, and friends [30,31]. Furthermore, bullying at work has significant emotional consequences and social harm (panic attack, depression, frustration, anxiety, anger, fear, or hostility) and physical consequences (stress, fatigue, headaches, sleep disorders, gastrointestinal problem, and cardiac problems) [32,33,34,35,36,37]. A previous study also showed that workplace bullying correlates with burnout at the workplace [38]. In addition, a Malaysian study revealed that there is a strong relationship between workplace bullying and neuroticism [39]. Both harassment and bullying encompass a traumatic experience and can reduce the quality of life of the victims [40]. Since the behavior of bullying can vary from obvious verbal or physical assaults to elusive psychological abuse, it can cause a range of psychological and physical illnesses among the victims, causing various impacts, such as anxiety and depression [41]. Furthermore, workplace bullying has also been linked to economic consequences, such as costs related to turnover, absenteeism, compensation, lost productivity, and insurance claims [42,43]. A British survey reported that workplace bullying led to the loss of over one million workdays. Furthermore, it has been reported that bullying resulted in a financial loss of up to 20 billion Euros per year. When staff is being bullied, those around the staff feel distracted by workplace stress and caused health problems. This situation increases the fiscal cost affected by bullying due to absenteeism and cost compensation. It is clear that the health effects on bullying victims have been proven to be expensive for the organization [44]. Additionally, the financial cost that involves the management time in addressing the case of bullying at work makes this cost even higher [41]. The link between stigma and bullying have been well-recorded. A study conducted in Pakistan shows that internalized (Hepatitis C Virus/HCV) stigma is positively associated with bullying at the workplace. The study also pointed out that stigma is a process through which workplace bullying impacts self-esteem. Hence, internalized stigma is linked to lower self-esteem among people with HCV in Pakistan [45]. Another study also reported an association between stigma and bullying by showing that 1/5th of children had experienced Acquired Immunodeficiency Syndrome (AIDS)-related stigma in South Africa. Sadly, this study also reported that 76.1% of AIDS orphans also experienced bullying [46]. Hence, it is important for organizations to have the ability to fight against workplace bullying in order to minimize these impacts. In order to get rid of workplace bullying, it is very crucial for every organization to set up and/or strengthen the regulation and legal policies by enhancing the commitment toward a work environment without bullying, harassment, and other violence. Several measures and strategies can be applied for dealing with workplace bullying, such as health and safety promotion, public awareness, proactive guide, a clear policy on bullying, management participation and involvement, and partnership [47].

### 1.4. Workplace Bullying Assessment

Recently, two methods were employed to assess workplace bullying. The first method is the subjective method [4,15], which is the identification of whether respondents are subject to bullying in the workplace. The second one is the operational method, which measures bullying from various types of negative acts for at least 6 months as experienced by the subjected person [4]. In addition, workplace bullying can be assessed using either direct or indirect measures. Direct measures consist of formal complaints about bullying, the Negative Acts Questionnaire-Revised (NAQ-R), Bullying Risk Assessment, Quine workplace bullying questionnaire, Obstetrics and Gynecology questionnaire, NHS Staff Survey, General Medical Council (GMC) National Training Survey (NTS), Trade Unions Professional Bodies and Charitable Organizations, and Witnessing bullying. The indirect measures include the General Health Questionnaire, sickness and absence levels, HSE Stress Management Standards Indicator Tool, exit interviews, and other measures [48].

One of the most widely utilized instrument is the Negative Acts Questionnaire-Revised (NAQ-R) [49,50]. The NAQ-R is a later tool version that was designed as an improvement of a previous instrument or original scale known as NAQ, which was proposed by the same researcher, Einarsen, to address weakness found in the NAQ concerning factor structure and some questionable and biased items [13,42,50]. The original scale consisted of 29 items that encompass personal- and work-related bullying, which was then reduced to 22 items in the revised version [7,51]. Three main aspects can be deciphered from this revised version, i.e., work-related bullying, person-related bullying, and physical intimidation [50]. The NAQ-R has been used in more than 100 further studies [3] in approximately 40 countries with different occupational settings [4,52]. The NAQ-R is available with no cost from a research team, the Bergen Bullying Research Group at University of Bergen, Norway. Hence, this tool has been used massively, particularly within European Countries [4]. The NAQ-R has been translated and tested in several previous studies into different languages according to the site of the study, including Arabic [3], Japanese [4], and Danish [53]. According to Einarsen et al. [50], the internal consistency for the 22 items is excellent (Cronbach’s alpha = 0.90).

### 1.5. Present Study

Indonesia, unfortunately, has not yet had a lot of statistical data and studies that identify the occurrence of bullying in the workplace. Nevertheless, the International Centre for Research on Women (ICRW) has reported that 84% of children in Indonesia have experienced violence, including bullying [54]. In general, bullying cases in Indonesia have been identified to be linked to the characteristics of body image, which can even lead to suicides. Unfortunately, bullying related to body shape and weight is difficult to avoid, as these traits are obvious and usually become the center of attention. For example, the physical differences of peoples’ bodies, especially obese people, are subjected to stigma and negative justification that can lead to bullying and, in some cases, even to suicide [55]. Another type of bullying that was discovered in Indonesia is the one associated to public health issues such as Human Immunodeficiency Virus (HIV), sexually transmitted infections, and unwanted pregnancies. According to Rutgers, the issues of harmful cultural stigmas and taboos regarding sexuality, especially among young people, change the perception and understanding of sexuality, and efforts to reduce the incidence of bullying and sexual abuse in Indonesia are required [56].

No specific data, however, have been reported regarding workplace bullying in this country. Nevertheless, the authors of this paper believe workplace bullying to be a pressing matter; for example, recently, a man in Indonesia killed his co-worker out of anger of being repeatedly ridiculed for being a fat person [57]. Interestingly, Indonesia has set up the regulation under Law No.1 of 1970 on occupational safety to encourage protections for all employees against incident and illness. The Article 86 (1) of Law No.13 of 2003 on Manpower points out that every worker has the right to receive occupational health and safety protection and to be protected against moral and psychosocial threats, as well as from threats to human dignity and religious values. This law declared that every human must be protected from violence, including bullying. Since there is a limited number of studies on workplace bullying in Indonesia, this study aimed to adapt the NAQ-R to the Indonesian context. Another purpose of this study was to assess the trend of bullying in the workplace in Indonesia. It is expected that this study would add to the existing knowledge on workplace bullying issues, especially important for government and institutions, to support measures to address workplace bullying in Indonesia.

The framework applied in this study was shown in Figure 1 below:

## 2. Materials and Methods

### 2.1. Subject Participants

This was a cross-sectional quantitative survey on employees from several sectors in Indonesia according to the authors’ network, such as construction, manufacturing, oil and gas, higher education, and health service sectors. A letter of confirmation with a proposal containing the information of the present study was sent to the targeted companies. An informed consent form was also distributed to participants to ensure them that no individual or company names would be reported or mentioned during data analysis and reporting. The study was carried out from May 2020 to November 2020. However, due to the COVID-19 pandemic, the survey was distributed through online questionnaires. Additionally, in relation to the COVID-19 situation, health protocols were applied in this study. Since all data collection were carried out online, the authors asked participants to follow the health protocol while completing the questionnaire in accordance with the Decree of the Minister of Health of the Republic of Indonesia No. HK.01.07/Menkes/328/2020 on the Guidelines for the Prevention and Control of Corona Virus Disease 2019 (COVID-19) in the Workplace to Support Business Continuity in a Pandemic Situation, which consisted of hand washing with soap or hand sanitizer, ensuring that the devices used for completing the questionnaire were clean, and keeping a minimum physical distance from other people of 1.5–2 m.

The inclusion criteria of this study were productive age (18 years old to >60 years old) and duration of working for at least 6 months. Meanwhile, the exclusion criteria were employed for less than 6 months, which followed the definition of workplace bullying [24]. A total of 4435 questionnaires were sent to through emails and a total of 3468 questionnaires were completed on the online system, giving a participation rate of 78.20%. After reviewing all data, 328 items were excluded due to uncompleted and missing data; thus, a total of 3140 (90.5%) respondents participated in this study. Ethical clearance was given by the research and community engagement ethical committee of the Faculty of Public Health, Universitas Indonesia, under the ethical approval letter number 583/UN2.F10.D11/PPM.00.02/2020.

### 2.2. Methods

On 11 May 2020, the authors received permission from Bergen Bullying Research Group to use the NAQ-R scale in this study. This tool was then translated into Indonesian using the instrument translation and adaptation process method. First, the English version was translated into Indonesian, which was then reviewed and modified by the authors. Afterward, the first translation was tested by five health experts, comprising a general practitioner, an occupational health and safety expert, a psychologist, and a counsellor. Then, this second version was back-translated into English to ensure the result was similar to the original version.

Two other instruments were also administrated in order to achieve the purpose of the present study, particularly the psychometric properties of Indonesian NAQ-R. These instruments were the K10 (psychosocial distress) scale [58] and the Satisfaction with Life Scale (SWLS) [59]. In addition, demographics information of the participants, such as gender, age, educational background, types of industry, absenteeism, and historical health status, was also collected. The hypothesis upheld in this study is that workplace bullying is positively associated with psychosocial distress and negatively linked to quality of life.

A trial assessment was conducted to assess the initial response for the final version of Indonesian NAQ-R and other tools used. A total of 90 respondents from occupational health and safety fields completed this validity assessment, but 23 respondents were excluded due to missing data; thus, 67 were included. The data analysis showed that the reliability (Cronbach’s Alpha) of the three instruments was excellent with values of 0.849, 0.869, and 0.758 for NAQ-R, K10, and SWLS respectively. One change was made on the NAQ-R, K10, and SWLC. Item 19 on the NAQ-R, regarding the “Pressure not to claim something which by right you are entitled to (e.g., sick leave, holiday entitlement, travel expenses)”, which was initially translated into Saya ditekan untuk tidak mengambil hak saya (misalnya cuti sakit, hak libur, biaya perjalanan), was changed into Saya tidak diperbolehkan untuk mengambil apa yang menjadi hak saya di tempat kerja (misalnya cuti sakit, hak libur, biaya perjalanan) or “I was not allowed to take something that I am entitled to in the workplace (e.g., sick leave, holiday entitlement, travel expenses)”. On the K10 scale, one question regarding “that everything was an effort”, in which “everything” was translated into “segalanya”, was modified into “semua yang diinginkan” or “everything that I want”. Furthermore, one question on SWLS was also modified. The item stating “If I could live my life over, I would change almost nothing”, which was translated into “Jika saya bisa mengulang hidup saya, saya tidak akan merubah apapun”, was modified into “Jika saya terlahir kembali, saya tidak akan merubah apapun dalam hidup saya” or “If I were reborn, I would not change anything in my life”.

After the trial assessment, the data collection was conducted in 11 companies from various industries that had agreed to participate in the study. These companies were from the oil and gas, construction, manufacturing, health services, and educational institution industries. The final stages of this study consisted of analyzing and disseminating the data (Figure 2).

### 2.3. Instrument

#### 2.3.1. Indonesian Version of NAQ-R

The NAQ-R has 22 items that assess the occurrence of bullying within the previous six months of work as experienced by the respondents. The respondents were asked to choose the response to the items that best describe the experience, which ranged from “Never” to “Now and then (occasionally)”, “Monthly”, “Weekly”, and “Daily”. Importantly, in order to avoid misunderstanding on the definition of bullying, each item in the instrument was phrased in behavioral terms to avoid the label of “Bullying or Harassment” that may confuse the participants [50]. However, there were three questions that were asked that used the term “bullying”. These questions were asked after the respondents were provided with the following definition of bullying: “We define bullying as a situation where one or several individuals persistently over a period of time perceive themselves to be on the receiving end of negative actions from one or several persons, in a situation where the target of bullying has difficulty in defending him or herself against these actions. We will not refer to a one-off incident as bullying”. The first question that was asked after the definition above was provided was whether the respondents had ever been bullied at work over the last six months. The respondents were asked to choose “No”, “Yes, but rarely”, “Yes, now and then”, “Yes, several times in a week”, or “Yes, almost daily”. A “Yes” response would categorize the respondents as a self-labeled workplace bullying victim [3,4,51,60]. The “Yes” answer would require the respondent to answer the second question, which was about the perpetrator (person who bullied) of the bullying that they experienced by giving a check mark on the applicable options. The options for the perpetrators were direct supervisor, other supervisor/manager, colleague, subordinate, customer, or other. The last question pertaining to the above definition was the number of persons who bullied them. The Indonesian version of NAQ-R can be shown in Appendix A.

#### 2.3.2. Psychosocial Distress

The Kessler 10 (K10) is an instrument used to assess how frequently the respondents experienced psychosocial distress in the past 30 days. The K10 is also used as a screening tool for mental health or psychosocial disorders through the 10 questions about the respondent’s feeling during the past month. Response categories are based on a five-point Likert scale ranging from never (0) to all of the time (4). The scores of 10 responses are then added up. A total score under 20 is categorized as “well” and a total score of 22–24 is categorized as “likely to have a mild mental disorder”. Moreover, total scores of 25 to 29 and 30 or above are interpreted as “likely to have moderate mental disorder” and “likely to have a severe mental disorder”, respectively [61,62]. The K10 scale has been used and translated into various languages, such as in Arabic, Chinese, Dutch, Hebrew, Italian, Japanese, Sinhalese, and Spanish [58]. The present study utilized the K10 scale translated into Indonesian (Appendix A).

#### 2.3.3. Satisfaction with Life Scale (SWLS)

The SWLS encompasses five items and was designed to assess the global judgment of person’s life satisfaction. This tool provides five statements where the respondents are asked to indicate whether they agree or disagree using a seven-point response category starting from 7 (strongly agree) to 1 (strongly disagree). The SWLS score can be interpreted to identify whether the respondent is satisfied or dissatisfied with their life. For instance, a score of 5–9 represents extreme dissatisfaction with life, while a score of 10–14 represent dissatisfaction with life. A score of 15–19 represents slight dissatisfaction; a score of 20 means neutral; a score of 21–24 represents slight satisfaction; and a score of 26–30 and 31–35 are indicative of being satisfied and extremely satisfied, respectively [59,63]. Since there are limited studies using SWLS in an Indonesian context, this tool was translated into the Indonesian language before being used in this study (Appendix A).

### 2.4. Statistical Analysis

A univariate analysis was performed to examine the differences in demographic characteristic of the respondents in this study. These characteristics included the variables of gender, age, educational background, types of industry, level of position, employment status, duration of working, history of illness, and absenteeism. In addition, internal consistency reliability was also assessed through Cronbach’s alpha coefficient. An Exploratory Factor Analysis (EFA) was used to test the structural validity of the 22 items in the Indonesian version of the NAQ-R extracting factor with eigenvalues of more than 1.0. In accordance with a previous study, there are three factors in the tool, namely work-related bullying, personal-related bullying, and physical intimidation [50]. Additionally, other studies reported three model factors: person- and work-related bullying, physical or psychological intimidation bullying, and occupational devaluation [4], while studies claiming two models have reported person- and work-related bullying [3]. We applied the KMO (Keiser-Meyer-Olkin) and Bartlett to test the assumption correlation between parameters. If the result shows that the KMO value is more than 0.5 and the *p*-value of Bartlett is less than 0.05, a correlation is established between the parameters that show that the factor analysis test can be continued. The present study tested three models: Model 1 (one factor model), Model 2 (two factor model), and Model 3 (three factor model). Furthermore, a confirmatory factor analysis test was conducted to examine the model fit by identifying the fit indices that consisted of the Comparative Fit Index (CFI), Root Mean Square Error of Approximation (RMSEA), Goodness of Fit Index (GFI), and Adjusted Goodness of Fit Index (ACGFI). The results could be considered to comply to the adequacy of model if the values of CFI, GFI, and ACGFI were higher than >0.90 and an RMSE score of less than 0.05. To determine the best model, the scores of Akaike Information Criteria (AIC) and Bayesian Information Criteria (BIC) were used. The smaller the AIC and BIC values were, the more appropriate the model to fit into the field condition. To examine the concurrent and constructive validity of the Indonesian version of NAQ-R, the Pearson correlation scores were calculated with other variables, such as the psychosocial distress and satisfaction with life. Additionally, the Mann–Whitney U test and Kruskal–Wallish H test were performed to assess the differences between the variables in the study. SPPS 24.0 (IBM Corporation, Armonk, NY, USA) and R Packages (R Foundation for Statistical Computing, Vienna, Austria) were used for data cleaning and analyses.

## 3. Results

### 3.1. Study Participants

A total of 3140 subjects participated in this study. The majority of the respondents was male (75.5%) as opposed to female (24.5%). Of all respondents, 25% were above 40 years old and came from various types of industry, including construction (32.2%), oil and gas (23.2%), and educational settings (11.2%). The characteristics of the participants are illustrated in Table 1.

### 3.2. Reliability Analysis of the Indonesian Version of NAQ-R

The internal consistency for the Indonesian version of the NAQ-R and other subscales in this study was presented in Table 2. The Cronbach’s alpha of NAQ-R was 0.897.

### 3.3. Factor Structure of the Indonesian Version of NAQ-R

The results of the assumption test showed that the KMO score was higher than 0.5 and the *p*-value of the Bartlett test was <0.05. It can be concluded that there was a correlation between each parameter and that an evaluation of the factor analysis can be performed. The exploratory factors that were yielded according to the present study (hereafter referred to as the Indonesian Model) involved a different item from the previously reported studies (Table 3). In addition, all factor loadings had a score of more than 0.3. Therefore, these parameters reflected each factor.

The result of the confirmatory factor analysis tested from the three distinct measurement models (according to the previous studies) was a marginal fit (GFI, AGFI, and CFI < 0.90 and RMSEA > 0.05). The GFI was 0.9, 0.84, and 0.88, for Model 1, Model 2, and Model 3, respectively, while the AGFI was 0.87, 0.87, and 0.86 for Model 1, Model 2, and Model 3, respectively. The CFI was 0.85, 0.84, and 0.84 for Model 1, Model 2, and Model 3, respectively, while the RMSE was 0.07, 0.07, and 0.07 for Model 1, Model 2, and Model 3, respectively. Vice versa, the GFI, AGFI, and CFI for the Indonesian Model show a good fit (>0.9), with values of 0.92, 0.91, and 0.90, respectively, while the RMSE showed a marginal fit with a score above 0.05 (0.06). Moreover, the AIC and BIC values show that the Indonesian Model was lower as opposed to the other models of 73,903.06 (AIC) and 74,187.51 (BIC). This means that in the present study, the model fits the Indonesian contexts.

### 3.4. Concurrent and Constructive Validity of Indonesian NAQ-R

As shown in Table 4, a significant and strong correlation exists between the score of the Indonesian NAQ-R and psychosocial distress, whereas a negative and the weakest relationships were revealed between the NAQ-R and satisfaction with life. It can be concluded that the higher the NAQ-R score is, the lower the satisfaction with life; thus, the higher the psychosocial distress score is, the lower the satisfaction with life.

An analysis of variance using the Mann–Whitney U test and Kruskal–Wallish H test on NAQ-R, psychosocial distress, satisfaction with life, and demographic variables was conducted. The result of this analysis revealed that NAQ-R and satisfaction with life did not have a significant difference in gender (U = 905,611; *p* = 0.753 and U = 891,140; *p* = 0.337), while others variables were significantly different (*p* =< 0.05).

### 3.5. The Prevalence of Workplace Bullying, Psychosocial Distress, and Satisfaction with Life

Table 5 summarizes the characteristics linked to bullying. It was identified that 89.2% of respondents had never experienced bullying. Moreover, 8.1% and 2.1% of participants reported as being bullied rarely and sometimes, respectively. According to the prevalence of bullying, the perpetrators were colleagues (8.5%), immediate superior (2.4%), and other superiors or managers in the organization (2.1), with the majority being male perpetrators (6.3%). In addition, 74% of respondents were likely to be well and 16% of respondents were likely to have a mild mental disorder (Figure 3). Regarding the trend of satisfaction with life, the majority of subjects in this study was satisfied (30%), and 18% felt slightly dissatisfied (Figure 4). Table 6 depicts the percentage of each item in the Indonesian version of NAQ-R.

## 4. Discussion

The aim of the present study was to examine the psychometric properties of the Indonesian translation of NAQ-R. This new version of the NAQ-R had an acceptable level of internal consistency reliability, with a Cronbach’s alpha ranging between 0.721 and 0.897. This coefficient is slightly below the previous study reported from the original version (0.90) [50], Greek version (0.915) [64], Spanish version (0.91) [65], Japanese version (0.91–0.95) [4], and Arabic version (0.63–0.90) [3]. However, the new version of NAQ-R seems to be reliable in the Indonesian context for measuring workplace bullying.

Based on the present study, the confirmatory factor analysis of 22 NAQ-R items proposed three different extracted factors with 11, seven, and four items respectively. This finding was different from previous studies [3,4,50]. It has been observed that our findings were comparable to the previous models, yet these previous models were not suitable for Indonesian contexts because, statistically, the CFI, GFI, and ACGFI scores were less than 0.90 and the RMSE score was higher than 0.05. In addition, since the AIC and BIC values of this present study showed the lowest score among the existing models, the Indonesian version is increasingly in line with the real conditions in the field. Therefore, a new model has been proposed in this study. The extracted Factor 1 was grouped as person-related bullying, where the original study identified Factor 1 as work-related bullying [50]. In addition, our finding of the first factor was also slightly different from those reported by studies from the Arabic version (work-related bullying) [3] and the Japanese version, where the factor consists of both person-related and work-related bullying [4]. Interestingly, our finding is slightly consistent with the Italian study, in that Factor 1 is revealed as person-related bullying [66]. Factor 2 consists of seven items that were labeled as work-related bullying, which is also similar to the Italian study. Meanwhile, other studies categorized this as person-related bullying [3,50] or physical or psychological intimation [4]. Factor 3 consisted of four items, which was named intimidation towards a person. Each item of this factor has an indication that efforts and performances of employees are not appreciated, for example, being the target of spontaneous anger, persistent criticisms of work and effort, and excessive monitoring of the employee’s tasks. Based on the degree of the severity or impact, factor 3 is the more severe kind of bullying. The Indonesian culture is unique and Indonesians have been classified as having a large power gap, having a weak uncertainty avoidance, being collectivist, and having a feminine culture [67]. Since there is no single Indonesian culture, Indonesian standard business culture is quite different. Importantly, in Indonesian culture, people are expected to have emotionally expressive lives. This means that it is difficult to control the emotional condition of colleagues at the workplace [68].

Concurrent and constructive validity of the Indonesian NAQ-R were examined with psychosocial distress and satisfaction with life. The present study revealed that workplace bullying measured by NAQ-R was positively linked to psychosocial distress and negatively correlated with life satisfaction. This finding is in line with studies conducted in Arab and Nigerian settings [3,69]. Several studies related to these associations have been documented. An Australian study showed that the occurrence of workplace bullying is more likely to trigger significant symptoms of depression in contrast with the experience of bullying at the workplace [70]. Moreover, it was also reported that absenteeism, poor health condition, sleep disorder, depressive symptoms, and diagnosis of depression is frequently identified among and correlated with employees with self-labeled bullying [71]. According to Malik and Björkqvist, there is a high correlation between workplace bullying and occupational stress in both male and female study participants [72], whereas a study by Kivima¨ki et al. proposed that workplace bullying foresees the onset of depression and long-time exposure to bullying is linked to higher risks of cardiovascular disease [73]. Importantly, exposure to workplace bullying has been considered to increase the risk of psychiatric, phycological, and psychosomatic problems [74].

On the one hand, experiencing bullying at the workplace could decrease job satisfaction; hence, it is important to control work-related stressors that could impact satisfaction with life among workers [75]. Bullying among workers can trigger an individual suffering in terms of a career progression, safety, self-esteem, and anxiety, causing life satisfaction issues [76]. Interestingly, bullying at the workplace can be conceptualized as the manifestation of stigma, which is caused by discrimination, especially for those facing situations that are socially seen to have devalued characteristic such as Hepatitis C, HIV, and people with leprosy; this stigma then affects self-esteem [77]. A previous study reported that workplace bullying is linked to low self-esteem because of internalized stigma [45]. Stigma has been presented as having five components, which include labeling and depersonalization, isolation, stereotyping, power, and denigration and reinforcement, which induce the discrimination of a person or peer group [2]. In fact, those who experience mental health issues can also be at risk of being bullied due to the stigma linked to mental health problems [78]. It was also pointed out that workplace bullying has detrimental impacts on job performance [79]. Employees with low levels of bullying showed better job performance as bullying negatively affects job satisfaction and turnover intention [80].

The present study shows that 10.8% of participants experienced bullying at their work environment with 8.1% being bullied rarely and 2.1%, 0.3%, and 0.2% being bullied now and then, several times in a week, and almost daily, respectively. This study is somewhat similar to a previous study that reported that 10.6% of victims were bullied rarely, while 0.5% were bullied on a daily basis [3]. In addition, a European study reported that 4.1% of participants reported exposure to bullying or harassment at work [81]. Moreover, a prospective follow-up study regarding self-labeled workplace bullying cited that 6.1% of employees reported being bullying now and then, and 1.4% experienced bullying at work on a daily to monthly basis [71]. A study in Cyprus reported that 45.6% of participants had been exposed to at least one bullying behavior at work [82]. Another finding stated that over 40% of respondents could identify experiencing bullying as causing depression symptoms [70]. In fact, the prevalence rate of workplace bullying has been captured in several countries, such as in Sweden (3.3%), Finland (16%), France (10.2%), Australia (15.2%), Norway (11%), and Belgium (8.3%) [83]. Our findings also pointed out that based on the participant’s responses, the perpetrators were dominated by both males than females. This result is in line with those of the previous studies. As far as the position level, colleagues tended to bully more than superiors. Contrarily, a previous study found that the perpetrators of bullying were mostly supervisors and managers [49]. From the victim’s perspective, our study showed no difference between males and females. Existing evidence has shown that bullying can occur at all times, affecting both men and women. In fact, gender is one of the fundamental variables in understanding the concept of bullying, particularly when observing the social characteristics of a community and describing the strong relationship between women and men regarding certain issues such as psychosocial distress. This study supported another scientific study by Niedhammer et al., where bullying at work was found to be a profound risk factor for depressive symptoms for both men and women [84].

Our study has as a strength that this is the first study to use the Indonesian version of the NAQ-R, which was confirmed to have acceptable levels of reliability, as well as concurrent and construct validity. Therefore, it will be useful as a tool in conducting surveys or further studies in Indonesia. Additionally, the sample size of the present study is quite big and adequate, strengthened by several methodological steps. However, several limitations have been identified. Firstly, due to the COVID-19 pandemic, the study was conducted online, which could affect the performance of respondents when completing the questionnaire. Moreover, several questions were not been completed properly and needed to be excluded from this study. Secondly, the instrument used is a self-questionnaire, and thus, based on employees’ perception, which could mean that the results are biased and subjective. Thus, further studies are needed to explore the information from participants. Thirdly, since the main purpose of the study is to test the validation and reliability of the Indonesian version of NAQ-R, this study did not examine in detail the multivariate analysis regarding the demographic variable and the prevalence of workplace bullying. In addition, the psychosocial distress and satisfaction with life variables are not explained in detail in the present study, as it focused on workplace bullying. Therefore, further studies are required in order to analyze and cover these limitations. We hypothesize that a thorough investigation into Indonesian culture as it pertains to bullying is needed to identify the specific connection between culture and workplace bullying. Future studies need to consider the heterogeneity of sample in other occupational settings and asses, in-depth, each item in order to understand the broader nature of workplace bullying.

The finding of the present study provides a comprehensive information, especially regarding the fact that this instrument can be used as an initial assessment to identify the prevalence of bullying at work. The authors believe that the knowledge of bullying can be useful for employers, particularly for health and safety experts, to understand how employees can comprehend the experience and the nature of bullying, as well as identifying its impact. It is clearly evident that bullying has a detrimental impact on both individuals and organizations. Hence, mitigation and coping strategies can be established in the workplace to reduce bullying-related incidents. Furthermore, with the addition of the reinforcement of the concepts of workplace bullying, organizations may seek to understand how bullying stems from social phenomena in the community. Importantly, bullying cannot be undermined or hushed up. Bullying and stigmatizing behaviors must be stopped and prevented. Therefore, active and intentional approaches should be applied to control and minimize bullying. The authors also implore the regulators to consider bullying aspects into policies that must be applied in companies to show the regulator’s commitment to address the issues of bullying.

## 5. Conclusions

This study underlines the psychometric properties, factor structure, and validity of the Indonesian version of NAQ-R. The nature and impacts of bullying have been clearly identified, as persistent exposure and prevalent problems in working life have detrimental impacts both on employees and organizations. It is crucial to recognize potential bullying in the workplace. Organizations should include bullying as one of the hazards or risks in their Occupational Health Safety Management System (OSHMS) as a part of mitigation and prevention controls to reduce the issues of workplace bullying. It has been concluded that the Indonesian version of NAQ-R developed in this study is a reliable and valid tool to assess workplace bullying. Thus, the Indonesian version of NAQ-R is a useful tool to be used as a screening system to identify the prevalence of workplace bullying that can reveal information regarding high-risk groups, risk factors, impacts, and so on.

## Figures and Tables

**Figure 1 ijerph-18-03985-f001:**
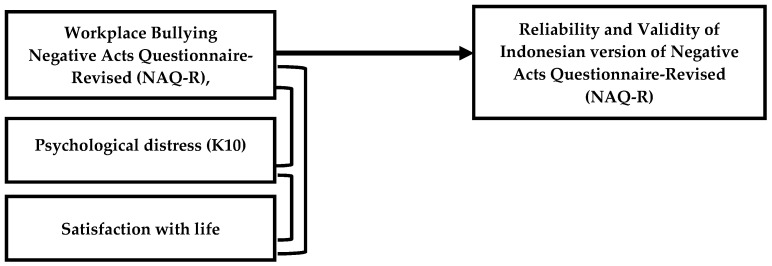
Study framework.

**Figure 2 ijerph-18-03985-f002:**
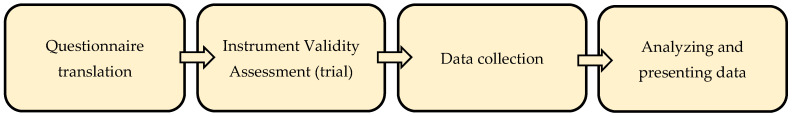
Study method.

**Figure 3 ijerph-18-03985-f003:**
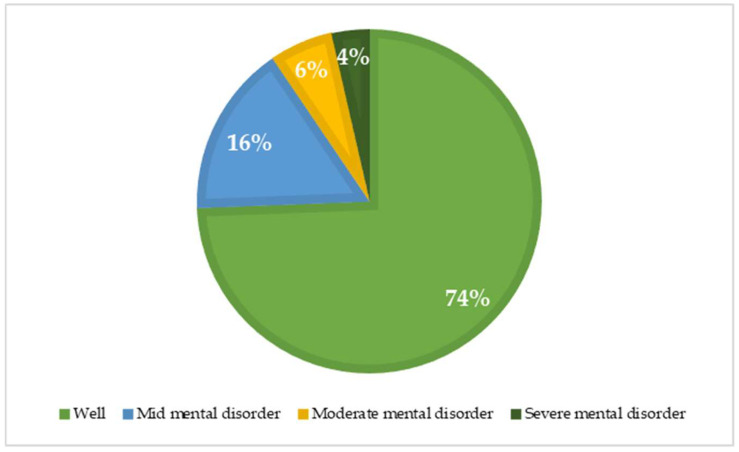
Prevalence of psychological distress.

**Figure 4 ijerph-18-03985-f004:**
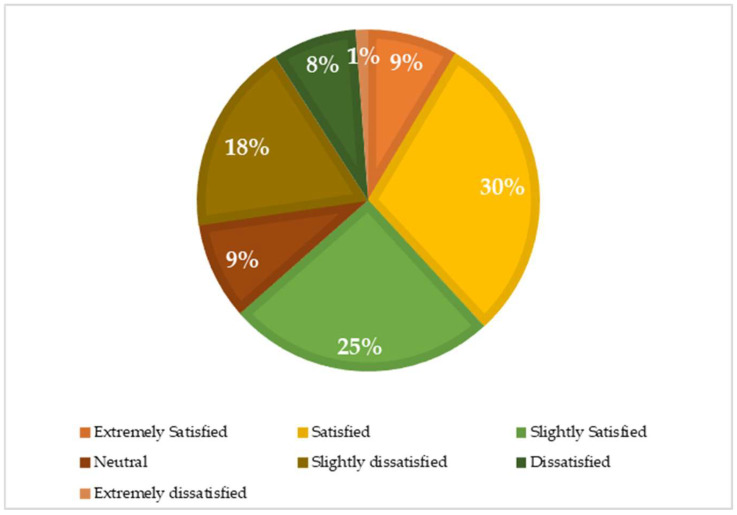
The prevalence of satisfaction with life.

**Table 1 ijerph-18-03985-t001:** Demographic characteristics (*n* = 3140).

Characteristics	*n* (%)
Gender	
Male	2370 (75.5)
Female	770 (24.5)
Age	
<25 years old	390 (12.4)
25–29 years old	784(25)
30–34 years old	607 (19.3)
35–40 years old	558 (17.8)
>40 years old	801 (25.5)
Educational Background	
Elementary School	77 (2.5)
Junior High School	141 (4.5)
Senior High School	1042 (33.2)
Diploma (D3)	365 (11.6)
Undergraduate (D4/S1)	1327 (42.3)
Master Program (S2)	185 (5.9)
Doctoral Program (S3)	3 (0.1)
Types of Industry	
Oil and Gas	727 (23.2)
Manufacturing	228 (7.3)
Construction	1011 (32.2)
Education	351 (11.2)
Health services	327 (10.4)
Call Centre	201 (6.4)
Power plant	220 (7)
Others	75 (2.2)
Level of Position	
Operator/Admin	968 (30.8)
Staff	922 (29.4)
Supervisor	327 (10.4)
Assistant Manager	97 (3.1)
Manager	167 (5.3)
Others	659 (21)
Employment Status	
Permanent employee	1267 (40.3)
Contract employee	1219 (38.8)
Outsourcing/third party employee	437 (13.9)
Daily	220 (7)
Duration of working	
<3 years	1398 (44.6)
4–6 years	487 (15.5)
7–10 years	518 (16.4)
>10 years	714 (22.7)
Minimum Wage	
Under Minimum Regional Wage (UMR)	303 (9.6)
Similar Minimum Regional Wage (SMR)	987 (31.4)
Higher than Minimum Regional Wage (HMR)	1850 (58.9)
History of illness (experience of chronic diseases such as diabetes, heart problems, stroke, osteoporosis, hypertension, etc.)	
Yes	212 (6.8)
No	2696 (85.9)
Unknown	225 (7.2)
Absenteeism (Due to illness)	
0 day	1785 (56.8)
1–5 days	1120 (35.7)
6–10 days	146 (4.6)
>10 days	89 (2.8)
Absenteeism (Due to non-illness)	
0 day	1460 (46.5)
1–5 days	1213 (38.6)
6–10 days	275 (8.8)
>10 days	192 (6.1)

**Table 2 ijerph-18-03985-t002:** Internal consistency of the 22-item Indonesian NAQ-R, K10, and SWLS.

Instrument	N	N Items	Cronbach’s (α)
NAQ-R Total	3140	22	0.897
Factor 1 (person-related bullying)	3140	11	0.860
Factor 2 (work-related bullying)	3140	7	0.777
Factor 3 (intimidation towards a person)	3140	4	0.721
Psychosocial Distress	3140	10	0.881
Satisfaction with life	3140	5	0.841

**Table 3 ijerph-18-03985-t003:** Exploratory factor analysis of NAQ-R.

Factor	Item	Item Wording *	Factor Loading
Factor 1 (person-related bullying)	2	Being humiliated or ridiculed in connection with your work (p) (pw’) (p”)	0.605
	5	Spreading of gossip and rumors about you (p) (pw’) (p”)	0.594
	6	Being ignored or excluded (being ‘sent to Coventry’) (p) (pw’) (p”)	0.634
	7	Having insulting or offensive remarks made about your person (i.e., habits and background), your attitudes, or your private life (p) (pw’) (p”)	0.716
	9	Intimidating behavior such as finger-pointing, invasion of personal space, shoving, or blocking/barring the way (i) (pw’) (p”)	0.530
	10	Hints or signals from others that you should quit your job (p) (pi’) (p”)	0.636
	12	Being ignored or facing a hostile reaction when you approach (p) (pw’) (p”)	0.583
	15	Practical jokes carried out by people you do not get on with (p) (pi’) (p”)	0.661
	17	Having allegations made against you (p) (pw’) (p”)	0.517
	20	Being the subject of excessive teasing and sarcasm (p) (pw’)	0.712
	22	Threats of violence or physical abuse or actual abuse (i) (pi’)	0.584
Factor 2 (work-related bullying)	1	Someone withholding information which affects your performance (w) (pw’) (w”)	0.515
	3	Being ordered to do work below your level of competence (w) (od’) (w”)	0.595
	4	Having key areas of responsibility removed or replaced with more trivial or unpleasant tasks (p) (od’) (w”)	0.603
	14	Having your opinions and views ignored (w) (pw’) (p”)	0.595
	16	Being given tasks with unreasonable or impossible targets or deadlines (w) (pw’) (w”)	0.657
	19	Pressure not to claim something which by right you are entitled to (e.g., sick leave, holiday entitlement, travel expenses) (w) (pw’)	0.474
	21	Being exposed to an unmanageable workload (w) (pw’) (w”)	0.639
Factor 3 (intimidation towards a person)	8	Being shouted at or being the target of spontaneous anger (or rage) (i) (pw’) (p”)	0.633
	11	Repeated reminders of your errors or mistakes (p) (pw’)	0.589
	13	Persistent criticism of your work and effort (p) (pw’) (p”)	0.663
	18	Excessive monitoring of your work (w) (pw’) (w”)	0.636

* (w)—work-related bullying, (p)—person-related bullying, (i)—physically intimidating bullying according to Einersen et al. [50]; (pw)—person- and work-related bulling, (pi)—physical or psychological intimidation bullying, (od)—occupational devaluation according to Tsuno et al. [4]; (w”)—work bullying, (p”)—personal bullying according to Makarem et al. [3].

**Table 4 ijerph-18-03985-t004:** Correlation between Indonesian NAQ-R, K10, and SWLS.

No	Item	Mean (SD)	5 Psychosocial Distress	6 Satisfaction with Life
1	NAQ-R Total	27.50 (6.43)	0.627	−0.242
2	Person-related bullying	12.72 (2.90)	0.515	−0.227
3	Work-related bullying	9.37 (2.69)	0.566	−0.163
4	Intimidation towards a person	5.41 (1.90)	0.505	−0.246
5	Psychosocial Distress	16.54 (5.77)	1	
6	Satisfaction with life	22.97 (6.15)	−0.307	1

All correlations are significant at the 0.001 level (two-tailed).

**Table 5 ijerph-18-03985-t005:** NAQ-R respondent characteristics.

Characteristics	*n* (%)
Bullied at work	
No	2801 (89.2)
Yes, but rarely	255 (8.1)
Yes, now and then	67 (2.1)
Yes, several times in a week	10 (0.3)
Yes, almost daily	7 (0.2)
Perpetrators	
Immediate superior	76 (2.4)
Other superiors/managers in the organization	67 (2.1)
Colleagues	266 (8.5)
Subordinates	27 (0.9)
Customers/patients/students, etc.	26 (0.8)
Others	16 (05)
The number and gender of perpetrators Male perpetrators	
None	2862 (91.1)
1–2 persons	197 (6.3)
3–4 persons	50 (1.6)
5–6 persons	21 (0.7)
>6 persons	10 (0.3)
Female perpetrators	
None	2983 (95)
1–2 persons	117 (3.7)
3–4 persons	22 (0.7)
5–6 persons	10 (0.3)
>6 persons	2 (0.0)

**Table 6 ijerph-18-03985-t006:** Percentage of each item of in the Indonesian NAQ-R (N = 3140).

Over the Last Six Months, How Often Have You Been Subjected to the Following Negative Acts at Work	Never (%)	Now and then (%)	Monthly (%)	Weekly (%)	Daily (%)
Someone withholding information which affects your performance	1989 (63.3)	1079 (34.4)	28 (0.9)	25 (0.8)	19 (0.6)
2. Being humiliated or ridiculed in connection with your work	2451 (78.1)	647 (20.6)	31 (1)	8 (0.3)	3 (0.1)
3. Being ordered to do work below your level of competence	1987 (63)	1306 (33)	52 (1.7)	24 (0.8)	50 (1.6)
4. Having key areas of responsibility removed or replaced with more trivial or unpleasant tasks	2525 (80.4)	555 (17.7)	24 (0.8)	21 (0.7)	15 (0.5)
5. Spreading of gossip and rumors about you	2107 (67.1)	968 (30.8)	27 (0.9)	17 (0.5)	21 (0.7)
6. Being ignored or excluded (being ‘sent to Coventry’)	2710 (86.3)	408 (13)	10 (0.3)	5 (0.2)	7 (0.2)
7. Having insulting or offensive remarks made about your person (i.e., habits and background), your attitudes, or your private life	2686 (85.5)	432 (13.8)	10 (0.3)	6 (0.2)	6 (0.2)
8. Being shouted at or being the target of spontaneous anger (or rage)	2464 (78.5)	620 (19.7)	31 (1)	22 (0.7)	3 (0.1)
9. Intimidating behavior such as finger-pointing, invasion of personal space, shoving, or blocking/barring the way	2948 (93.9)	175 (5.6)	9 (0.3)	4 (0.1)	4 (0.1)
10. Hints or signals from others that you should quit your job	2834 (90.3)	281 (8.9)	12 (0.4)	6 (0.2)	7 (0.2)
11. Repeated reminders of your errors or mistakes	1927 (61.4)	1028 (32.7)	83 (2.6)	53 (1.7)	49 (1.6)
12. Being ignored or facing a hostile reaction when you approach	2602 (82.9)	521 (16.6)	9 (0.3)	4 (0.1)	4 (0.1)
13. Persistent criticism of your work and effort	2034 (64.8)	985 (31.4)	70 (2.2)	34 (1.1)	17 (0.5)
14. Having your opinions and views ignored	1748 (55.7)	1336 (42.5)	30 (1)	9 (0.3)	17 (0.5)
15. Practical jokes carried out by people you do not get on with	2560 (81.5)	551 (17.5)	15 (0.5)	5 (0.2)	9 (0.3)
16. Being given tasks with unreasonable or impossible targets or deadlines	2030 (64.6)	950 (30.3)	91 (2.9)	46 (1.5)	23 (0.7)
17. Having allegations made against you	2844 (90.6)	280 (8.9)	10 (0.3)	3 (0.1)	3 (0.1)
18. Excessive monitoring of your work	2515 (80.1)	510 (16.2)	58 (1.8)	26 (0.8)	31 (1)
19. Pressure not to claim something which by right you are entitled to (i.e., sick leave, holiday entitlement, travel expenses)	2797 (89.1)	315 (10)	21 (0.7)	3 (0.1)	4 (0.1)
20. Being the subject of excessive teasing and sarcasm	2885 (91.9)	243 (7.7)	8 (0.3)	1 (0.0)	3 (0.1)
21. Being exposed to an unmanageable workload	2454 (78.2)	630 (20.1)	37 (1.2)	9 (0.3)	10 (0.3)
22. Threats of violence or physical abuse or actual abuse	2977 (94.8)	153 (4.9)	7 (0.2)	3 (0.1)	0 (0.0)

## Data Availability

The data presented in this study are available on request from the corresponding author.

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
