# Peer review of "Identification of Workplace Bullying: Reliability and Validity of Indonesian Version of the Negative Acts Questionnaire-Revised (NAQ-R)"

_ijerph, 2021, doi:10.3390/ijerph18083985_

Round 1

Reviewer 1 Report

Comments for the Author: 

This manuscript reports a Identification of Workplace Bullying: Reliability and Validity of Indonesian Version of the Negative Acts Questionnaire-Revised (NAQ-R). I believe this manuscript provide us very unique and new ideas. But still there are some issues in this manuscript. Although I found the manuscript is very interesting and authors present the unique idea of the present paper. I found myself struggling to get past a number of rather significant limitations linked to the comments I have outlined below. I hope you will find this helpful in improving your manuscript.

The main concern is related to the fact that the abstract part is too long. Please just provide the proper justification and just talk about to the point in the abstract part. Don't need to write a long abstract . Further, authors must have to put the all questionnaires in the appendix. Authors must have to provide the proper findings in the abstract part. The contributions and research gap of the present paper is very clear.

Introduction:

Indonesia, unfortunately, have not yet had a lot of statistical data and studies that identify the occurrence of bullying in the workplace. Nevertheless, the International Centre for Research on Women (ICRW) has reported that 84% children in Indonesia have experienced violence, including bullying. I believe there are limited studies on the workplace bullying from Indonesia. Does workplace bullying effect on the health and safety? Does bullying is highly associated with anger, sadness, fear, or hostility? Further, both harassment and bullying encompass a traumatic experience and can reduce the quality of life of the victims. How could this happens? Harassment and bullying create a post traumatic stress on quality of the life?  Workplace  bullying has also been linked to economic consequences such as costs related to turnover,  absenteeism, compensation, lost productivity. Does workplace bullying significantly facilitate to turnover,  absenteeism, compensation, and lost productivity, if so, how could this happen? My main concern is related to that how we can get rid from workplace bullying? And what coping strategy would be right for that?  

Methodology:         

Authors used a cross-sectional quantitative survey on employees from several sectors according to the authors’ network such as construction, manufacturing, oil and gas, higher  education, and health service sectors in Indonesia. Authors tells us what is the criteria of the data collection from different Indonesian sector. Overall, method part is fine.

Discussion:

Please Give the proper headings of discussion, future research, and implications. 

Author Response

Responses to Reviewer 1

Comments for the Author:  

This manuscript reports a Identification of Workplace Bullying: Reliability and Validity of Indonesian Version of the Negative Acts Questionnaire-Revised (NAQ-R). I believe this manuscript provide us very unique and new ideas. But still there are some issues in this manuscript. Although I found the manuscript is very interesting and authors present the unique idea of the present paper. I found myself struggling to get past a number of rather significant limitations linked to the comments I have outlined below. I hope you will find this helpful in improving your manuscript.

Response:

Dear Sir/Madame

Thank you so much for your time to review our manuscript. All your feedback helps us to improve the quality of it, and thank you for all advices. Herewith, we updated according the outlined give below.

The main concern is related to the fact that the abstract part is too long. Please just provide the proper justification and just talk about to the point in the abstract part. Don't need to write a long abstract . Further, authors must have to put the all questionnaires in the appendix. Authors must have to provide the proper findings in the abstract part. The contributions and research gap of the present paper is very clear.

Response:

Thank you for suggestion. The abstract had been revised  and we also put the all questionnaires in appendix.

Introduction:

Indonesia, unfortunately, have not yet had a lot of statistical data and studies that identify the occurrence of bullying in the workplace. Nevertheless, the International Centre for Research on Women (ICRW) has reported that 84% children in Indonesia have experienced violence, including bullying. I believe there are limited studies on the workplace bullying from Indonesia. Does workplace bullying effect on the health and safety? Does bullying is highly associated with anger, sadness, fear, or hostility? Further, both harassment and bullying encompass a traumatic experience and can reduce the quality of life of the victims. How could this happens? Harassment and bullying create a post-traumatic stress on quality of the life?  Workplace  bullying has also been linked to economic consequences such as costs related to turnover,  absenteeism, compensation, lost productivity. Does workplace bullying significantly facilitate to turnover,  absenteeism, compensation, and lost productivity, if so, how could this happen? My main concern is related to that how we can get rid from workplace bullying? And what coping strategy would be right for that? 

Response: thank you for the concern. Does workplace bullying effect on the health and safety? Yes, it is very impact on health and safety, particularly the psychosocial aspects. in its recent development, bullying has become one of the concerns in the Occupational Health and Safety (OHS) field. Bullying can happen to anyone in any workplace, regardless of rank or income-level, but it is more prevalent in particular professions such as education and healthcare.

Does bullying is highly associated with anger, sadness, fear, or hostility? Further, both harassment and bullying encompass a traumatic experience and can reduce the quality of life of the victims. How could this happens? Harassment and bullying create a post-traumatic stress on quality of the life?

We added several information to support your questions.

Bullying is a very damaging and dangerous complex and heterogeneous phenomenon that directly affects hundreds of million dwellers per year. Understanding the definition of bullying has been proven to be massively useful as a starting point for research[1]. A most common definition of bullying is a repetition of a negative physical, verbal or psychological action targeted to a certain individual that apparently seen in both organization and community. It has been said that one of the origins of bullying is lessened community cohesion or destruction of  the warp and weft of the tapestries of society[2] (Line 23-30).

In addition, a Malaysian study revealed that there is a strong relationship between workplace bullying and neuroticism [39]. Both harassment and bullying encompass a traumatic experience and can reduce the quality of life of the victims [40]. Since behavior of bullying can vary from obvious verbal or physical assaults to elusive psychological abuse, it can cause a range of psychological and physical illnesses among the victims, causing various impact such as anxiety and depression [41].

Harassment and bullying create a post-traumatic stress on quality of the life?  Workplace  bullying has also been linked to economic consequences such as costs related to turnover,  absenteeism, compensation, lost productivity. Does workplace bullying significantly facilitate to turnover,  absenteeism, compensation, and lost productivity, if so, how could this happen?

Furthermore, workplace bullying has also been linked to economic consequences such as costs related to turnover, absenteeism, compensation, lost productivity, and insurance claims [42], [43].  A British survey reported that workplace bullying led to the loss of over one million workdays. It is also further reported that bullying resulted in a financial loss of up to 20 billion Euros per year. When staff is being bullied, those around the staff feels distracted with workplace stress and affected their health problems. This situation increases the fiscal cost affected by bullying due to the absenteeism and cost compensation. It is clearly prominent that the health effects on bullying victims has proven to be expensive for the organization [44]. Also, financial cost that involves the management time in addressing the case of bullying at work makes this cost even higher [41].

My main concern is related to that how we can get rid from workplace bullying? And what coping strategy would be right for that? 

Hence, it is important for organization to have the ability to fight against workplace bullying in order to minimize these impacts. In order to get rid of workplace bullying, it is very crucial for every organization to set up and or strengthen the regulation and legal policies by enhancing the commitment toward work in an environment without bullying, harassment and other violence. Several measures and strategies can be applied for dealing with workplace bullying such as health and safety promotion, public awareness, proactive guide, a clear policy on bullying, management participation and involvement, and partnership [47].

Methodology:          

Authors used a cross-sectional quantitative survey on employees from several sectors according to the authors’ network such as construction, manufacturing, oil and gas, higher  education, and health service sectors in Indonesia. Authors tells us what is the criteria of the data collection from different Indonesian sector. Overall, method part is fine.

Response:

Thank you for good comment.

Discussion: 

Please Give the proper headings of discussion, future research, and implications. 

Response: The finding of the present study provides a comprehensive information, especially regarding the fact that this instrument can be used as initial assessment to identify the prevalence of bullying at work. The authors believe that the knowledge of bullying can be useful for employers, particularly for the health and safety experts to understand how employees can comprehend the experience and the nature of bullying, as well as identifying its impact. It is clearly evident that bullying has a detrimental impact both on individuals and organization. Hence, mitigation and coping strategies can be established in the workplace to reduce bullying-related incidents. Further, with the addition of reinforcement of the concepts of workplace bullying, the organization may seek to understand how bullying stems from the social phenomena in the community. Importantly, bullying cannot be undermined or hushed up. Bullying and stigmatizing behaviours must be stopped and prevented. Therefore, active and intentional approaches should be applied to control and minimize bullying. The authors also believe that this study implies the regulators to consider the bullying aspects into policies that must be applied in companies to show the regulator’s commitment to address the issues of bullying.

Reviewer 2 Report

please find enclosed my comments on the manuscript “Identification of Workplace Bullying: Reliability and Validity 2 of Indonesian Version of the Negative Acts Questionnaire-Re-3 vised (NAQ-R)", which was submitted to the International Journal of Environmental Research and Public Health. My general impression after reading this manuscript is that this article is suitable, when it comes to the content and format, for publication in the IJERPH. However, it rarely happens at this stage  but I recommend only  minor revision, especially when it comes to better/more clear description of particular parts of this manuscript. Below I summarized my comments in respect to the specific parts of this manuscript:

Abstract – the abstract is too long and should be more concise to present clear: objective, method, results and conclusion. Please eliminate some statistical coefficients from the abstract, as they should be only mentioned in the results section.

Introduction – the authors did generally good job in discussing the issue of workplace bullying, its prevalence and assessment. However, the first paragraph of the introduction is lacking in literature review on bullying concept and its theoretical perspectives. The authors cite such definition of bullying:  “the situation refers to every hostile and  aggressive actions subjected to at least one or more victims with stigma”, but they did not expand this definition. I would recommend to elaborate more on the significance of stigma and bullying. Furthermore, the authors wrote ” Bullying tends to use references to certain characteristics of an individual such as race, religion, gender, sexual-orientation, and disability” – but again, they did not provide any examples or studies on that topic. Thus, please cite and discuss following papers, which deals with the link between stigma and bullying and present the situation of sensitive samples (general and clinical), which may be exposed to bullying and discrimination due to various aspects of social stigmatization (please take particularly notice to gender issues in this aspect):

Cluver, L., Orkin, M. (2009). Cumulative risk and AIDS-orphanhood: Interactions of stigma, bullying and poverty on child mental health in South Africa. Social Science & Medicine, 69, 1186-1193,

https://doi.org/10.1016/j.socscimed.2009.07.03

Huggins, M. (2016). Stigma Is the Origin of Bullying. Journal of Catholic Education, 19 (3). http://dx.doi.org/10.15365/joce.1903092016

Noor, A, Bashir S, Earnshaw VA. (2016). Bullying, internalized hepatitis (Hepatitis C virus) stigma, and self-esteem: Does spirituality curtail the relationship in the workplace. Journal of Health Psychology. 2016;21(9):1860-1869. doi:10.1177/1359105314567211

Finally, I was wondering if the authors could elaborate more on the Indonesia perspective – whether, and if so what it is unique in case of bullying and/or stigma in this country compared to studies in other parts of the world?

Method and Results – this part is very well presented.

Discussion – this part, like introduction, is poorly grounded in the classical literature on bullying and stigma. Above papers to cite and discuss also here will be helpful.

Taking everything into an account, I recommend publishing this manuscript in the IJERPH. However, it needs before some minor revisions, which I highlighted above.

Author Response

Comments and Suggestions for Authors

please find enclosed my comments on the manuscript “Identification of Workplace Bullying: Reliability and Validity 2 of Indonesian Version of the Negative Acts Questionnaire-Re-3 vised (NAQ-R)", which was submitted to the International Journal of Environmental Research and Public Health. My general impression after reading this manuscript is that this article is suitable, when it comes to the content and format, for publication in the IJERPH. However, it rarely happens at this stage  but I recommend only  minor revision, especially when it comes to better/more clear description of particular parts of this manuscript. Below I summarized my comments in respect to the specific parts of this manuscript:

Response: First of all, we would like thank you for your time and providing the feedbacks for our manuscripts. It was really helpful to improve its quality.

Abstract – the abstract is too long and should be more concise to present clear: objective, method, results and conclusion. Please eliminate some statistical coefficients from the abstract, as they should be only mentioned in the results section.

Response: The abstract had been revised and made it shorten.

Introduction – the authors did generally good job in discussing the issue of workplace bullying, its prevalence and assessment. However, the first paragraph of the introduction is lacking in literature review on bullying concept and its theoretical perspectives. The authors cite such definition of bullying:  “the situation refers to every hostile and  aggressive actions subjected to at least one or more victims with stigma”, but they did not expand this definition. I would recommend to elaborate more on the significance of stigma and bullying. Furthermore, the authors wrote ” Bullying tends to use references to certain characteristics of an individual such as race, religion, gender, sexual-orientation, and disability” – but again, they did not provide any examples or studies on that topic. Thus, please cite and discuss following papers, which deals with the link between stigma and bullying and present the situation of sensitive samples (general and clinical), which may be exposed to bullying and discrimination due to various aspects of social stigmatization (please take particularly notice to gender issues in this aspect):

Cluver, L., Orkin, M. (2009). Cumulative risk and AIDS-orphanhood: Interactions of stigma, bullying and poverty on child mental health in South Africa. Social Science & Medicine, 69, 1186-1193,

https://doi.org/10.1016/j.socscimed.2009.07.03

Huggins, M. (2016). Stigma Is the Origin of Bullying. Journal of Catholic Education, 19 (3). http://dx.doi.org/10.15365/joce.1903092016

Noor, A, Bashir S, Earnshaw VA. (2016). Bullying, internalized hepatitis (Hepatitis C virus) stigma, and self-esteem: Does spirituality curtail the relationship in the workplace. Journal of Health Psychology. 2016;21(9):1860-1869. doi:10.1177/1359105314567211

Response: thank you for great advice and providing the additional references. We tried to add some information according your recommendation above.

Bullying is a very damaging and dangerous complex and heterogeneous phenomenon that directly affects hundreds of million dwellers per year. Understanding the definition of bullying has been proven to be massively useful as a starting point for research[1]. A most common definition of bullying is a repetition of a negative physical, verbal or psychological action targeted to a certain individual that apparently seen in both organization and community. It has been said that one of the origins of bullying is lessened community cohesion or destruction of  the warp and weft of the tapestries of society[2].

Stigma is defined as an insidious process that create inappropriate shame and unworthiness to the individual. Stigma can lead to  bullying. When someone is being stigmatized, he or she will effectively cut off from the collective group. Moreover, bullies automatically deny access to resources, stability, protection and social status for the stigmatized person. They set boundaries to prevent social movements from the person who is stigmatized, and the only membership available for this person would be in the outside group, where the he or she may also be rejected due to the stigmatization [2].

Finally, I was wondering if the authors could elaborate more on the Indonesia perspective – whether, and if so what it is unique in case of bullying and/or stigma in this country compared to studies in other parts of the world?

Response: we added the information. In general, bullying cases in Indonesia have been identified to be linked to the characteristics of body image that even lead to decision of suicides. Unfortunately, bullying related to the body shape and weight is difficult to avoid as these traits are obvious and usually become the center of attraction. For example, physical differences in the body, especially in obese people, are followed by stigma and negative justification that can lead to a bullying and,  in some cases, even to suicide decisions [55]. Another type of bullying discovered in Indonesia is the one associated to public health issues such as Human Immunodeficiency Virus (HIV), sexually transmitted infections and unwanted pregnancies. According to Rutgers, the issues of harmful cultural stigmas and taboos regarding sexuality, especially for young people, change the perceptions and understanding of sexuality, and efforts to reduce the incidence of bullying and sexual abuse in Indonesia are really required [56].

Method and Results – this part is very well presented.

Response: Thank you for feedback.

Discussion – this part, like introduction, is poorly grounded in the classical literature on bullying and stigma. Above papers to cite and discuss also here will be helpful.

Response: Thank you for the advice. We added some information according to your advice.

On one hand, experiencing of bullying at workplace could decrease job satisfaction; hence, it is important to control work-related stressors that could impact satisfaction with life among workers [75]. Bullying among workers can trigger an individual suffering in terms of a career progression, safety, self-esteem, anxiety causing life satisfaction issues [76]. Interestingly, bullying at workplace can be conceptualized as manifestation of stigma, which is closed to discrimination, especially for those facing situations that are socially seen as devalued characteristic such as Hepatitis C, HIV, people with leprosy, and then this stigma has affected self-esteem [77]. A previous study reported the workplace bullying is linked with low self-esteem by internalized stigma [45]. Stigma has been presented as having five components that include labelling and depersonalization; isolation; stereotyping; power; denigration and reinforcement which induce the discrimination of a person or peer group [2]. In fact, those who experience mental health issues can also be at risk of being bullied due to the stigma linked to mental health problems [78].

Taking everything into an account, I recommend publishing this manuscript in the IJERPH. However, it needs before some minor revisions, which I highlighted above.

Reviewer 3 Report

Overall:

  • There are numerous formatting and grammatical errors within the body of the manuscript and the tables that need to be corrected.
  • Authors need to ensure that they are using the correct country abbreviation for the United States of America – specifically on line 72.
  • What is the framework that guided this research?

Introduction:

  • The authors need to be more succinct description of bullying. From line 37 to line 46 the information could be condensed and place in the authors on words. This section does not read smoothly.
  • On line 47, the authors state that bullying and harassment are different but they do not show the contrast between the two words.
  • Line 65, the authors state “Three models of workplace bullying have been clearly described”; however, there was not clear description within the manuscript or an in-text citation for the descriptions.
  • The paragraph starting “Investigations have been performed on the prevalence” has references to the United Kingdom bookending a reference to the United States. The preference would be to keep the UK references together.
  • Lines77-78, “The prevalence in Scandinavian and European is 3.5-10%” – Scandinavian and European what?? Also is the 3.5-10% in reference to both Scandinavian and European? Additionally, if referring to the countries of these particular regions, Scandinavian countries are considered European countries and thus this is very confusing.
  • Lines78-79, “while the prevalence in New Zealand and Australia is 79 18%[23] and 50% [24] and 25% [25],” – there are 3 percentages listed with only 2 countries listed. Needs to be corrected?
  • The entire paragraph under the subheading “1.3. Impact of Workplace Bullying” (Lines 84-99) are very repetitive and needs to be completely rewritten.

Materials and Methods:

  • Line 173 – The number of completed questionnaires is listed, however, there is no reference to the number of invitations that were sent out to complete the questionnaire. Thus there is no participation rate (%).
  • Line 193, Hypothesis should be written in present tense not past tense.
  • Lines 197-199, the following statement needs to be reworded – “A total of 90 respondents from occupational health and safety field took part in this validity assessment but only 67 completed the survey with the remaining (n=23) was excluded due to missing data.” Rewording needs to state that 90 were completed but only 67 were included.
  • Line 228, how is this defined “Now and then” and was that definition given to the participants? This does not seem consistent with the other response descriptions.
  • Why did the authors reference K6 and seem to make it the focal point of the paragraph beginning at Line 249 when they utilized K10, not K6?

Results:

  • What is the justification for the age categories in Table 1?
  • What is the difference between contract employee and outsourcing categories in Table 1?
  • The variable “History of Illness” and the response categories in Table 1 is very vague. Explain justification for the wording of this variable.
  • Explain the labels used in the columns labeled 5 & 6 in Table 4.
  • Definitions are needed for the categories of psychological distress in Figure 2.

Discussion:

  • Lines 380-381, there needs to be an explanation of why the models from previous studies would “not fit in Indonesian contexts.”
  • Lines 383-387, why did the authors choose to compare their finds with the Arabic, Japanese, and Italian studies.
  • Line 447, “This study supported other scientific study by Niedhammer et al.,” how specifically, other than just showing no difference between the genders from the victim perspective.
  • Limitation – not a generalizable study outside of Indonesia

Author Response

Comments and Suggestions for Authors

Overall:

  • There are numerous formatting and grammatical errors within the body of the manuscript and the tables that need to be corrected.

Response: Thank you for great feedback. We have already reviewed and corrected some errors in our manuscript.

  • Authors need to ensure that they are using the correct country abbreviation for the United States of America – specifically on line 72.

Response: we updated it into the United States of America.

  • What is the framework that guided this research?

Response: the study framework has been updated in figure 1.

Introduction:

  • The authors need to be more succinct description of bullying. From line 37 to line 46 the information could be condensed and place in the authors on words. This section does not read smoothly.

Response: Thank you for advice, we added the sentences In addition, several other definitions of bullying have been proposed by researchers inclusing stalking or psychosocial measures which is not always visible.

  • On line 47, the authors state that bullying and harassment are different but they do not show the contrast between the two words.

Response: Thank you, we updated it according to your recommendation

Bullying and harassment are extremely similar that they are often utilized conversely to represent harmful or negative behaviours to other persons. It, however, differs in terms of definition. Bullying tends to be prompted by a hidden personal characteristic of target or victims, such competence, popularity or integrity. For example, work-related bullying may take the form of making unreasonable demands and taking credit for others’ work. In contrast, harassment refers to unwanted, offensive and intrusive behaviors related to sexual, racial or physical elements. In other words, harassment tends to use references to certain characteristics of an outward individual such as race, religion, gender, sexual-orientation, and disability and it has a string physical component such as contact and damage to possessions in a person’s work [11].

  • Line 65, the authors state “Three models of workplace bullying have been clearly described”; however, there was not clear description within the manuscript or an in-text citation for the descriptions.

Response: we added the information. Three models of workplace bullying have been clearly described. The first is work-related bullying where a person withholds information that can affect performance. The second is person-related bullying, for instance actions related to persistent criticism of errors or mistakes and sarcasm. The third is is physically intimidating bullying such as intimidating behaviours (finger-pointing, shoving, or blocking victim’s way).

  • The paragraph starting “Investigations have been performed on the prevalence” has references to the United Kingdom bookending a reference to the United States. The preference would be to keep the UK references together.

Response: Thank you for the advice, it has been updated Investigations have been performed on the prevalence of workplace bullying around the world.  In the United Kingdom, for instance, 10.6% of respondents in a random nationwide survey of 70 organizations are identified as bullying victims. According to a study by UNISON (the Public Service Union in the UK), workplace bullying has been identified in the public sector union with an incidence of 34% [23].

  • Lines77-78, “The prevalence in Scandinavian and European is 3.5-10%” – Scandinavian and European what?? Also is the 3.5-10% in reference to both Scandinavian and European? Additionally, if referring to the countries of these particular regions, Scandinavian countries are considered European countries and thus this is very confusing.

Response: It revised into The prevalence in European countries ranges between 3.5 and 10% [24].

  • Lines78-79, “while the prevalence in New Zealand and Australia is 79 18%[23] and 50% [24] and 25% [25],” – there are 3 percentages listed with only 2 countries listed. Needs to be corrected?

Response: Thank you for the correction. It has been corrected. while the prevalence in New Zealand and Australia is 18%[25] and 25% to 50% [26,27] respectively.

  • The entire paragraph under the subheading “1.3. Impact of Workplace Bullying” (Lines 84-99) are very repetitive and needs to be completely rewritten.

Response: Thank you for the feedback. We have already reviewed and updated this section. Effect of bullying on several aspects of life is now well-established. The impact of workplace bullying includes negative impacts on the bullying target as individual, as well as on the organization, family, and friends [30],[31]. Further, bullying at work has significant emotional consequences and social harms (panic attack, depression, and frustration, anxiety, anger, fear, or hostility) and physical consequences (stress, fatigue, headaches, sleep disorders, gastrointestinal problem, and cardiac problems) [32–37]. A previous study also shows that workplace bullying correlates with burnout at workplace [38]. In addition, a Malaysian study revealed that there is a strong relationship between workplace bullying and neuroticism [39]. Both harassment and bullying encompass a traumatic experience and can reduce the quality of life of the victims [40]. Since behavior of bullying can vary from obvious verbal or physical assaults to elusive psychological abuse, it can cause a range of psychological and physical illnesses among the victims, causing various impact such as anxiety and depression [41]. Furthermore, workplace bullying has also been linked to economic consequences such as costs related to turnover, absenteeism, compensation, lost productivity, and insurance claims [42], [43].  A British survey reported that workplace bullying led to the loss of over one million workdays. It is also further reported that bullying resulted in a financial loss of up to 20 billion Euros per year. When staff is being bullied, those around the staff feels distracted with workplace stress and affected their health problems. This situation increases the fiscal cost affected by bullying due to the absenteeism and cost compensation. It is clearly prominent that the health effects on bullying victims has proven to be expensive for the organization [44]. Also, financial cost that involves the management time in addressing the case of bullying at work makes this cost even higher [41]. The link between stigma and bullying have been well-recorded. A study conducted in Pakistan shows that internalized (Hepatitis C Virus/HCV) stigma is positively associated with bullying at workplace. The study also pointed out that stigma is a process through which workplace bullying impacts self-esteem. Hence, internalized stigma is linked to lower self-esteem among people with HCV in Pakistan [45]. Another study also reported an association between stigma and bullying by showing that 1/5 of children identified with the experience of Acquired Immunodeficiency Syndrome (AIDS)-related stigma in South AfricaSadly, this study also reported that 76.1% of AIDS-orphans also experience bully [46]. Hence, it is important for organization to have the ability to fight against workplace bullying in order to minimize these impacts. In order to get rid of workplace bullying, it is very crucial for every organization to set up and or strengthen the regulation and legal policies by enhancing the commitment toward work in an environment without bullying, harassment and other violence. Several measures and strategies can be applied for dealing with workplace bullying such as health and safety promotion, public awareness, proactive guide, a clear policy on bullying, management participation and involvement, and partnership [47].

Materials and Methods:

  • Line 173 – The number of completed questionnaires is listed, however, there is no reference to the number of invitations that were sent out to complete the questionnaire. Thus there is no participation rate (%).

Response: we added the information 4,435 questionnaires were sent to through emails and a total of 3,468 questionnaires were completed on the online system, giving a participation rate of 78.20%.

  • Line 193, Hypothesis should be written in present tense not past tense.

Response: it revised to present tense: The hypothesis uphold in this study is that workplace bullying is positively associated with psychosocial distress and negatively linked to the quality of life.

  • Lines 197-199, the following statement needs to be reworded – “A total of 90 respondents from occupational health and safety field took part in this validity assessment but only 67 completed the survey with the remaining (n=23) was excluded due to missing data.” Rewording needs to state that 90 were completed but only 67 were included.

Response: we updated into A total of 90 respondents from occupational health and safety field completed in this validity assessment, but only 67 were included and the remaining (n=23) was excluded due to missing data.

  • Line 228, how is this defined “Now and then” and was that definition given to the participants? This does not seem consistent with the other response descriptions.

Response: Thank you for the feedback. We refers “now and then” according to the original paper or instrument. Yet, we added the information the definition of now and then with from time to time “Now and then (occasionally)”.

  • Why did the authors reference K6 and seem to make it the focal point of the paragraph beginning at Line 249 when they utilized K10, not K6?

Response: we revised it focusing on K10. The Kessler 10 (K10) is an instrument used to assess how frequently the respondents experienced psychosocial distress in the past 30 days. The K10 is also used as a screening tool for mental health or psychosocial disorders through the 10 questions about the respondent’s feeling during the past month. Response categories are based on a 5-point Likert scale ranging from none of the time (0) to all of the time (4). The scores of 10 responses are then added up. A total score under 20 is categorized as  “well” and a total score of 22-24 is categorized as “likely to have a mild mental disorder”, Meanwhile total scores of 25 to 29 and 30 or above are interpreted as “likely to have moderate mental disorder” and “likely to have a severe mental disorder”, respectively [61],[62]. K10 scale has been used and translated into various languages, such as in Arabic, Chinese, Dutch, Hebrew, Italian, Japanese, Sinhalese, and Spanish [58]. The present study utilized the K10 scale which was translated into Indonesian.

Results:

  • What is the justification for the age categories in Table 1?

Response: Thank you for the feedback. We considered the age according to Indonesian central bureau which group age of workforce. But, we comprised in 5 categories.

  • What is the difference between contract employee and outsourcing categories in Table 1?

Response: Contract employee is under the company with employment agreement, while the outsourcing is based on third party employee.

  • The variable “History of Illness” and the response categories in Table 1 is very vague. Explain justification for the wording of this variable.

Response: History of illness refers to experience of chronic diseases such as diabetes, heart problems, stroke, osteoporosis, hypertension etc.

  • Explain the labels used in the columns labeled 5 & 6 in Table 4.

Response: it revised according to your recommendation

  • Definitions are needed for the categories of psychological distress in Figure 2.

Response: Thank you for the advice. We apologize that we did not mention the categories earlier . Now, we have already updated it. The scores of 10 responses then are added up. In total, score under 20 are categorized as to be well, score 22-24 are likely to have a mild mental disorder, while score 25 to 29 are likely to have moderate mental disorder and score 30 and more are like to have a severe mental disorder

Discussion:

  • Lines 380-381, there needs to be an explanation of why the models from previous studies would “not fit in Indonesian contexts.”

Response: Thank you for concern. After the statistical test analysis, the models from previous studies would not fit in Indonesian contexts. We added the information. because statistically the CFI, GFI, and ACGFI scores were less than 0.90 and an RMSE score was higher than 0.05. In addition, since the AIC and BIC values of this present study showed the lowest score as opposed to previous models it means that Indonesian version is increasingly in accordance with the real conditions in the field.

  • Lines 383-387, why did the authors choose to compare their finds with the Arabic, Japanese, and Italian studies.

Response: According to the literature review, we found the NAQ-R had been translated into those language (Arabic, Japanese, and Italian). Hence, we considered those finding to our present study.

  • Line 447, “This study supported other scientific study by Niedhammer et al.,” how specifically, other than just showing no difference between the genders from the victim perspective.

Response: we added the information. from the victim perspective, our study shows no difference between males and females. Existing evidence has convinced researchers that bullying can occur at all the times to both men and women. In fact, gender is one of the fundamental variables in understanding the concept of bullying, particularlywhen observing the social characteristics of a community and describing the strong relationship between female and male regarding certain issues such as psychosocial distress. This study supported other scientific study by Niedhammer et al., that bullying at work was found to be profoundly risk factor for depressive symptoms for both men and women [85].

  • Limitation – not a generalizable study outside of Indonesia

Response: Thank you for the feedback. We agree and we tried to not being a generalizable study outside of Indonesia.

Round 2

Reviewer 3 Report

Thank you for the updated to your manuscript.